# Neurohumoral Markers of Cardiac Autonomic Denervation after Surgical Ablation of Long-Standing Persistent Atrial Fibrillation

**DOI:** 10.3390/life13061340

**Published:** 2023-06-07

**Authors:** Alexey Evtushenko, Vladimir Evtushenko, Anna Gusakova, Tatiana Suslova, Yulia Varlamova, Konstantin Zavadovskiy, Denis Lebedev, Anton Kutikhin, Elena Pavlyukova, Sergey Mamchur

**Affiliations:** 1Department of Cardiovascular Surgery, Research Institute for Complex Issues of Cardiovascular Diseases, 6 Sosnovy Boulevard, Kemerovo 650002, Russia; ave@kemcardio.ru (A.E.); mamchse@kemcardio.ru (S.M.); 2Cardiology Research Institute, Tomsk National Research Medical Center, Russian Academy of Sciences, 111a Kievskaya Street, Tomsk 634012, Russia

**Keywords:** persistent atrial fibrillation, autonomic nervous system, norepinephrine, radiofrequency ablation

## Abstract

Although the autonomic nervous system has an evident impact on cardiac electrophysiology and radiofrequency ablation (RFA) is the conventional technique for treating persistent atrial fibrillation, the specific effects of RFA have been insufficiently studied to date. Here, we investigated whether RFA affects neurohumoral transmitter levels and myocardial ^123^I-metaiodobenzylguanidine (^123^I-MIBG) uptake. To perform this task, we compared two groups of patients with acquired valvular heart disease: patients who had undergone surgical AF ablation and patients with sinus rhythm. The decrease in norepinephrine (NE) level in the coronary sinus had a direct association with the heart-to-mediastinum ratio (*p* = 0.02) and a negative correlation with ^123^I-MIBG uptake defects (*p* = 0.01). The NE level decreased significantly after the main surgery, both in patients with AF (*p* = 0.0098) and sinus rhythm (*p* = 0.0039). Furthermore, the intraoperative difference between the norepinephrine levels in the ascending aorta and coronary sinus (ΔNE) of –400 pg/mL was determined as a cut-off value to evaluate RFA efficacy, as denervation failed in all patients with ΔNE < –400 pg/mL. Hence, ΔNE can be utilized to predict the efficacy of the “MAZE-IV” procedure and to assess the risk of AF recurrence after RFA.

## 1. Introduction

As has been established over the years, the autonomic nervous system (ANS) plays a significant role in modulating the cardiac electrophysiology and arrhythmogenesis through its effect on cardiac physiology [1]. Cardiovascular disease may have an adverse prognostic tendency if the ANS components are dysfunctional, which could be a prognostic factor. The ANS has been the subject of numerous studies over the years which have contributed to a better understanding of its anatomy and physiology. Additionally, studies have shown the connection between ANS dysfunction and clinically significant arrhythmias, suggesting that the ANS might be a therapeutic target in arrhythmology [1].

A significant discovery was made with regard to neurohumoral markers that were found to be reflective of autonomic activity in the heart. Unfortunately, there is no versatile method for studying the ANS, and current techniques in this field impose certain requirements that are not always possible to meet. By identifying specific ANS markers and triggers, the study of autonomic cardiac activity can become more efficient than many other invasive methods. Patients with long-term persistent atrial fibrillation (AF) might benefit most from this.

AF represents a typical and common complication of cardiac surgery in 30% of cases of coronary artery bypass surgery, 40% of patients after valvular heart surgery and 50% of cases if these interventions are combined [2,3]. The conventional approach to the treatment of AF in patients with mitral valve disease is a one-stage mitral valve correction and radiofrequency procedure “MAZE” [4,5,6,7]. However, the effectiveness of such treatment is limited and allows the recovery of the sinus rhythm in 70–80% of patients, often being associated with structural, functional, and electrophysiological changes in the heart [8,9]. The risk of postoperative AF recurrence can reach 100% if there is a past medical history of AF [10]. Postoperative AF is considered to be an independent predictor of many adverse outcomes, including a two- to four-fold increased risk of stroke, bleeding, infection, renal or respiratory failure, cardiac arrest, cerebral complications, and the need to implant a permanent pacemaker. AF is associated with a two-fold higher case fatality rate in the postoperative period. Although the exact pathogenesis of developing postoperative AF requires further investigations, recent studies highlight the contribution of the interactions between pre-existing reasons behind the AF as well as systemic inflammation [11,12,13]. Postoperative AF is associated with multiple risk factors, including advanced age and severe comorbidities which are responsible for cardiac remodeling and vascular events. The clinical management of AF includes both preventive and therapeutic approaches, though their effectiveness is disputable. The postoperative period for patients with AF that have been insufficiently treated with RFA is associated with a high risk of recurrence. Evidently, the need for a clear definition of the AF treatment endpoint is crucial for preventing recurrence [14,15,16].

Normal cardiac electrophysiology is, to a significant extent, governed by the ANS. Pre- and postganglionic sympathetic and parasympathetic fibers form a complex network and synapses on extrinsic and intrinsic cardiac ganglia. Both sympathetic and parasympathetic fibers directly innervate the cardiac myocytes. One clinically significant implication is that the evaluation of the ANS modulation of the myocardium following RFA can be useful in predicting the recurrence of AF, and other adverse cardiac events as well [17,18,19]. An alteration in the autonomic tone of the heart may have adverse effects on the local cellular electrophysiology of the heart, which can manifest clinically in a variety of ways, ranging from a slowing of the heart rate to changes in the rate of heart contractions. The cardiac ANS plays a pivotal role in the onset and maintenance of AF, though a number of complex and disputed mechanisms exist.

The role of sympathovagal imbalance in the pathogenesis of arrhythmias is still a serious and unresolved problem for practitioners and researchers. The development of this direction has been propelled by Pappone and colleagues who showed a relationship between the impact on the paraganglionic nerve plexi of the heart and AF recurrence after catheter ablation [20]. Their destruction on the open heart was performed by Doll, in 2008 [21]. Anatomical ablation of the autonomic ganglionic plexi of the heart, proposed by Pokushalov and colleagues [22], has been suggested as an effective option to treat AF, but there are no methods for objectively evaluating the results of these interventions today.

It is therefore reasonable to speculate that denervation may be a common mechanism underlying many of the therapeutic effects associated with various procedures for treating AF. However, the effects of denervation on the long-term efficiency of AF surgical treatment require further studies. As research on the relationships between autonomic tone and cardiac dysrhythmias continues to evolve, there is increasing evidence that autonomic ganglia play an essential role in the pathogenesis of AF.

To obtain information about the state of the autonomic innervation of the heart, the determination of heart rate variability and measurements of epinephrine and norepinephrine (NE) concentrations in blood plasma are widely used [23,24]. However, these methods provide only indirect information about the predominance of the tone of the sympathetic or parasympathetic parts of the ANS, and it is impossible to assess the rhythm variability in patients with AF [25,26].

To date, the best way of carrying out the visual and quantitative assessment of sympathetic innervation of the myocardium is the use of nuclear medicine methods such as positron emission tomography (PET) of the heart with labeled catecholamines [27,28]. However, even though PET offers an exceptionally high diagnostic capability, there are certain limitations when it comes to its widespread use in clinical practice. Therefore, scintigraphy with ^123I^-metaiodobenzylguanidine (^123I^-MIBG) has acquired the greatest popularity in assessing the sympathetic activity of the myocardium today [29,30,31,32,33].

Given the lack of a “gold standard” assessment of the sympathetic innervation of the heart in vivo, in the present study, myocardial scintigraphy with ^123^I-MIBG was compared with the results of the quantitative measurements of metanephrine, normetanephrine, and NE in plasma obtained intraoperatively from the coronary sinus and ascending aorta.

The aim of this study was to evaluate the consequences of RFA on neurohumoral transmitter levels and ^123^I-MIBG uptake in patients with long-standing persistent AF.

## 2. Materials and Methods

The study was conducted according to the Good Clinical Practice guidelines and the latest revision of the Declaration of Helsinki (2013). The study protocol was approved by the Local Ethical Committee of the Research Institute for Complex Issues of Cardiovascular Diseases (Kemerovo, Russia, protocol code 2021/11/03, date of approval: 3 November 2021) and Cardiology Research Institute within Tomsk National Research Medical Centre (Tomsk, Russia, protocol code TNRMC-134/1, date of approval: 23 November 2021). Written informed consent was provided by all study participants after receiving a full explanation of the study’s purposes. Comorbid conditions (arterial hypertension, chronic heart failure, chronic obstructive pulmonary disease, asthma, chronic kidney disease, diabetes mellitus, overweight, and obesity) were diagnosed and treated according to the corresponding guidelines of the respective research societies. Clinicopathological information was collected at the time of hospital admission. In total, we enrolled 68 patients with mitral valve disease recommended for surgical correction: those suffering from long-standing persistent AF (*n* = 53) and those with sinus rhythm (SR, *n* = 15, Table 1).

The criterion of inclusion was the preserved function of the sinus node (SN) determined by intraoperative electrophysiological examination (Figure 1). Among the criteria of exclusion were comorbid conditions, pericardial adhesions, multivessel coronary artery disease, and multiple organ failure. Primary endpoints were cardiovascular death and major adverse cardiovascular events: myocardial infarction, stroke, and AF recurrence. Secondary endpoints were pacemaker implantation and reduced atrial mechanical function.

The patients were given 7 days prior to surgery to discontinue medications that affect the rhythm and heart conduction. During the intraoperative electrophysiological examination, we measured sinus node recovery time (SNRT), corrected sinus node recovery time (CSNRT), and Wenckebach point. Patients with preserved SN function were enrolled and randomized into two groups using envelopes. In the first group of patients, RFA was administered in combination with postganglionic plexus ablation based on the Doll (2008) scheme, while in the second group, lesions were performed according to the MAZE-IV scheme without the addition of postganglionic plexus ablation. In the case of normal SN function, the procedure was performed using the MAZE-IV technique (penetrating technique) to seal atrial appendages and additional postganglionic plexi ablation. Patients with SR who underwent surgery only due to valvular lesions comprised the control group (*n* = 15).

Each patient underwent a standard set of clinical and laboratory tests, including complete blood count (Sysmex XN-550, Kobe, Japan), biochemical profiling using an automated biochemical analyzer (Konelab 60i, Thermo Fisher Scientific, Waltham, MA, USA), urinalysis, X-ray examination (GE Healthcare TMX R+, General Electric Healthcare, Chicago, IL, USA), echocardiography (Sonos 2500 Diagnostic Ultrasound System, Hewlett Packard, Palo Alto, CA, USA), color duplex screening (Vivid 7 Dimension Ultrasound System, General Electric Healthcare, Chicago, IL, USA), spirometry (Microlab Spirometer, Vyaire Medical, Chicago, IL, USA), and coronary angiography (Innova 3100 Cardiac Angiography System, General Electric Healthcare, Chicago, IL, USA) in accordance with the guidelines [34]. Planar scintigraphy with ^123^I-metaiodbenzylguanidine (^123^I-MIBG) was performed before and after the surgery. ^123^I-MIBG is an analogue of guanethidine adrenergic blocker which allows the visualization of sympathetic innervation of the heart and assessment of myocardial adrenergic nerve activity [35]. During the myocardial scintigraphy, we assessed heart-to-mediastinum ratio, early and late radiopharmaceutical (RP) washout, and RP uptake defect. Functional class of heart failure was defined by a 6 min walking test.

Patients with persistent AF (group I) differed from patients with SR (group II) with regard to left atrial (LA) diameter and results of 6 min walking test. Chronic rheumatic heart disease was diagnosed in 50% of patients with persistent AF and in 44.4% of patients with SR. Connective tissue dysplasia was found in 28 and 44.4% of cases, respectively. Some patients from both groups required coronary artery bypass grafting (1–2 grafts).

Surgical treatment was performed in accordance with the recommendations for the management of patients with valvular heart diseases (ESC/EACTS, 2017). The frequency of mitral valve (MV) repair with annuloplasty rings in the groups was 50 and 55.5%, respectively, and MV prosthetic valve replacement was performed in one-third of cases in both groups. Several patients required multi-valvular interventions (for instance, single-stage tricuspid valve repair was performed in 22–28% of cases), and 4 patients underwent simultaneous intervention on the aortic valve. In the case of paroxysmal AF, an antiarrhythmic therapy was prescribed in group I.

Blood samples were collected intraoperatively from the ascending aorta and coronary sinus prior to cardiopulmonary bypass and 10 min after removal of the clamp from the aorta. The study was performed in the absence of any medications that could affect the catecholamine levels in the samples. In the case of any catecholamine injections, patients were excluded from the study. NE level in blood was assessed both before and after RFA. For the quantitative determination of plasma NE level, we applied Noradrenalin enzyme-linked immunosorbent assay kit (RE59261, IBL, Hamburg, Germany) according to the manufacturer’s protocol. Colorimetric analysis was conducted using Multiskan Sky microplate spectrophotometer (Thermo Fisher Scientific, Waltham, MA, USA). In order to avoid any external influence on the sympathetic tone of the heart during surgery, sympathomimetic drugs were withdrawn before the main phase of the operation, during and after the main stage before the sampling of blood from the ascending aorta and the coronary sinus. All patients underwent 24 h Holter ECG monitoring before the surgery and after removal of sutures before hospital discharge. A re-examination was performed 6 months after the operation.

Statistical analysis was performed using STATISTICA 13 software (TIBCO Software, Palo Alto, CA, USA). For descriptive statistics, data were presented as median (Me), upper and lower quartiles (Uq and Lq, respectively). Depending on the data distribution, two independent groups were compared using Student’s *t*-test or Mann–Whitney U test. To assess the significance of intragroup differences (measurements before and after the exposure within one group), Wilcoxon matched pairs signed rank test was used. *p*-values ≤ 0.05 were regarded as statistically significant. Discriminant analysis was used to build a model for the assessment of the RFA efficacy based on neurohumoral factors.

## 3. Results

In patients with long-standing persistent AF, the median heart-to-mediastinum ratio in the early and late phase of the study before the intervention did not differ from the patients with SR (1.75 (1.59; 1.91) and 2.06 (1.7; 2.1), respectively, *p* = 0.1). The washout rate did not show statistically significant differences either (24.5 (10.8; 41.1) % and 24.3 (12.3; 29.9) %, respectively, *p* = 0.15). The ^123^I-MIBG uptake defect before surgery was comparable between the groups (13.0 (7.0; 24.0) % and 19.0 (9.5; 23.5%) %, respectively, *p* = 0.59). At the pre-operative stage, the parameters of the sympathetic nervous system tonus were similar across the groups. Four weeks after the surgery, patients underwent a postoperative examination (Table 2).

The efficacy of AF RFA was evaluated using postoperative ^123^I-MIBG scintigraphy with the concurrent measurement of metanephrine, normetanephrine, and NE in the blood withdrawn from the ascending aorta and coronary sinus. In patients with atrial RF fragmentation (group I), the heart-to-mediastinum ratio after intervention in the late phase of the study was significantly lower compared to the patients with SR: (1.5 (1.4; 1.6) and 1.8 (1.56; 1.83), respectively, *p* = 0.02), whilst the uptake defect was significantly higher (25.0 (24.0; 35.0) % and 15.0 (12.0; 20.0) %, respectively, *p* = 0.01). A statistically significant (*p* = 0.01) decrease in the NE level in patients with RFA was found in the blood collected from the coronary sinus after the main stage of the surgery as compared to the preoperative level. In contrast, no significant difference (*p* = 0.2) in the NE level was found in the control group.

Further analysis demonstrated a significant intergroup difference (*p* = 0.004) in the postoperative NE level (Figure 2). A decrease in the heart-to-mediastinum ratio directly correlated with the reduction in NE level in the coronary sinus of patients with RFA (group I) compared to the control group with SR, whilst an increased RP uptake defect inversely correlated with the reduced NE level in the coronary sinus. The differences between the subgroups with the “MAZE-IV” procedure in the heart-to-mediastinum ratio (Figure 3), the RP washout rate (Figure 4) and the RF uptake defect (Figure 5) were statistically insignificant.

With the aim of building a model for the proper assessment of RFA efficacy based on neurohumoral factors, we applied a discriminant analysis and calculated the Wilks’ lambda, a statistic which evaluates the discriminatory ability of the function (i.e., the contribution of each independent variable into the model). The significance of the intraoperative difference between the levels of NE, metanephrine, and normetanephrine in the ascending aorta and in the coronary sinus (ΔNE, ΔNM, and ΔNNM, respectively) was measured by the F-value. If the F-value exceeded the critical value (1.23), the variable was kept in the model. In the case of ΔNE, the F-value was 13.20 and was retained in the model as its *p*-value was 0.001, in contrast to ΔNM and ΔNNM where the F-value also slightly exceeded 1.23 (1.41 and 1.34, respectively) but the *p*-value did not reach 0.05 (0.24 and 0.25, respectively). Wilks’ lambda was 0.56, with *p* = 0.0037, indicating the adequacy of the created model (Table 3).

ΔNE was therefore considered as a statistically significant factor. The calculation of ΔNE for each patient was performed as follows:(1)$$\Delta \mathrm{NE}={\mathrm{NE}}_{\mathrm{AA}}-{\mathrm{NE}}_{\mathrm{CS}},$$
where NE is for norepinephrine, AA is for ascending aorta, and CS is for coronary sinus.

As the percentage of correct assignments (i.e., the predictions of favorable or adverse outcome) was 88.9%, we concluded the model had high discriminative power in terms of its sensitivity/specificity (Table 4).

Upon the calculation of the coefficients for the canonical linear discriminant function and plotting of ΔNE values across the RFA outcomes, we found that all patients with ΔNE < −400 pg/mL suffered from AF paroxysms in the early postoperative period (indicative of the perfect sensitivity of this cut-off) whereas, at higher ΔNE values, only one case of AF paroxysm was detected (≈5% of such patients, suggestive of good specificity). Hence, we propose a ΔNE of –0.400 pg/mL as a cut-off reflecting RFA efficacy (Figure 6).

As described above, we detected a reduced NE level and the accumulation of ^123^I-MIBG in all patients who underwent RFA of postganglionic nerve plexi, and these parameters were associated with the effectiveness of the AF surgical treatment.

## 4. Discussion

AF is a frequent heart rhythm disorder, the etiology of which is still not fully understood [36,37,38,39,40], although it is generally accepted that sympathovagal imbalance can lead to paroxysmal supraventricular arrhythmia [36] and paroxysmal AF [41]. Therefore, local denervation of the heart can be an efficient treatment modality when applied to patients with long-standing persistent AF [42,43,44,45].

The effect of the ANS sympathetic component on cardiac electrophysiology is complex and depends on myocardial function. The predominance of sympathetic activity in the intact heart leads to a shortening of the cardiac action potential duration and to a decrease in the repolarization dispersion. In the injured myocardium, sympathetic activity leads to an increase in the repolarization dispersion and to the early afterdepolarization generation [1,46] providing a proarrhythmogenic effect. Further, studies based on the analysis of heart rate variability have shown that AF paroxysm can be the result of the combined activation of the sympathetic and parasympathetic components [47,48,49,50]. Sympathetic activity is the cause of increased calcium transit, whereas an increase in the parasympathetic tonus leads to the shortening of the atrial effective refractory period. The difference between the calcium transit and the AP duration (normally interdependent) augments Na^+^/Ca^++^ pump activity, which is responsible for the generation of early afterdepolarization and trigger activity [1,47].

The cardiac ANS comprises the extrinsic and intrinsic innervation of the heart. The postganglionic neurons of the heart may still function normally despite active central regulatory mechanisms. Studies by Choi and colleagues demonstrated a connection between the external and internal mechanisms of sympathetic heart tonus activation. Though activation of extracardiac ANS contributed to the onset of most AF paroxysms, in some cases (11%, as estimated by Choi and colleagues) the internal nervous activity of the heart preceded the external one and led to the development of AF paroxysms [41]. In addition, the heart’s internal sympathetic activity can activate the external components of the ANS, causing negative consequences. This mechanism is the basis of AF and it is implemented by sympathovagal imbalance [41]. In this sense, modulating other ANS elements is highly likely to control heart rate.

Among the most common approaches for studying ANS is the analysis of heart rate variability, although it has several limitations. First, this method reflects only the sympathovagal balance between the ANS components but not their activity. Second, this analysis requires an intact SN that mediates an adequate cardiac response to ANS activity, whereas a notable proportion of patients with persistent AF and heart failure have comorbid SN dysfunction. This imposes restrictions on using heart rate variability for studying the ANS tonus and this is not suitable for some patient categories [1]. An objective assessment of the ANS requires quantitative methods such as ^123^I-MIBG scintigraphy, yet they still lack clear diagnostic criteria [24]. Another important limitation of ^123^I-MIBG is that its current use is limited to specialized clinics and research centers due to the specificity of this isotope. Other methods are not sufficiently specific to identify cardiac autonomous activity, or they take an excessive amount of time, which restricts their application. Thus, there is no versatile method to measure the ANS activity of the heart [23].

This study shows that neurohumoral markers, such as NE, detectable by cheap and broadly employed enzyme-linked immunosorbent assay, can serve as a quantitative measure of the effectiveness of RFA in long-term persistent AF. Importantly, low concentrations of NE and its short decay period permit its measurement only in the absence of its entry into the blood from other sources.

The difference in the NE level in the blood collected from the aorta and the coronary sinus after the main stage of the surgery (ΔNE) is an informative factor in assessing the effectiveness of autonomic denervation of the heart, as ΔNE < −400 pg/mL can be considered as a reliable marker of unsuccessful denervation. As the effectiveness of AF surgical treatment was confirmed by 24 h Holter ECG monitoring, we then developed a mathematical model to evaluate the quality of atrial RF denervation based on the Wilks’ lambda, which makes possible the prediction of AF recurrence in the postoperative period. These data correlated with the scintigraphic and clinical data, allowing us to state that the completeness of heart autonomic denervation directly correlates with the effectiveness of the MAZE-IV RF procedure.

Considering a number of factors influencing the recurrence of postoperative AF, a decrease in the neurohumoral activity of the ANS enables the SR to be maintained and reduces the risk of postoperative complications, such as a reduction in left ventricular ejection fraction and systemic thromboembolism. The development of a tool to predict arrhythmia-associated complications has been the cornerstone of this study. An important result is the comparison of scintigraphic and laboratory methods for studying the autonomic activity of the heart, which will enable further research for new criteria and technologies.

The limitations of this study include the relatively small number of patients, the need to cancel the medications affecting ANS activity, the ability to take blood samples only in an “open” heart, and the limited options for administering sympathomimetics during the surgery. Although the results of the study showed the importance of determining the NE level in the assessment of autonomic atrial denervation, further studies are necessary to obtain more data.

## 5. Conclusions

Neurohumoral markers of the ANS in the postoperative period may indicate the efficiency of the RFA and MAZE-IV RF procedure, as the intraoperative gradient between the levels of NE in the ascending aorta and coronary sinus (ΔNE) of −400 pg/mL discriminates well between a successful and an unsuccessful RFA. Hence, we propose ΔNE as a promising tool for assessing the atrial denervation quality, although replication studies are clearly needed.

## Figures and Tables

**Figure 1 life-13-01340-f001:**
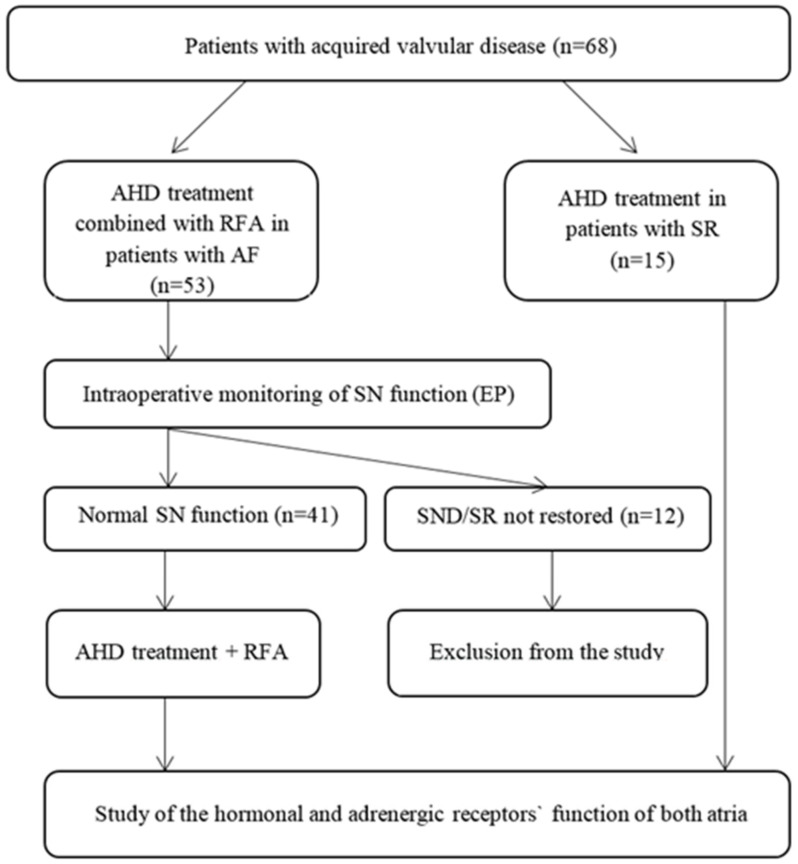
The design of the study. Note: AHD—acquired heart diseases; RFA—radiofrequency ablation; SR—sinus rhythm; SN—sinus node; EP—electrophysiological study; SND—sinus node dysfunction.

**Figure 2 life-13-01340-f002:**
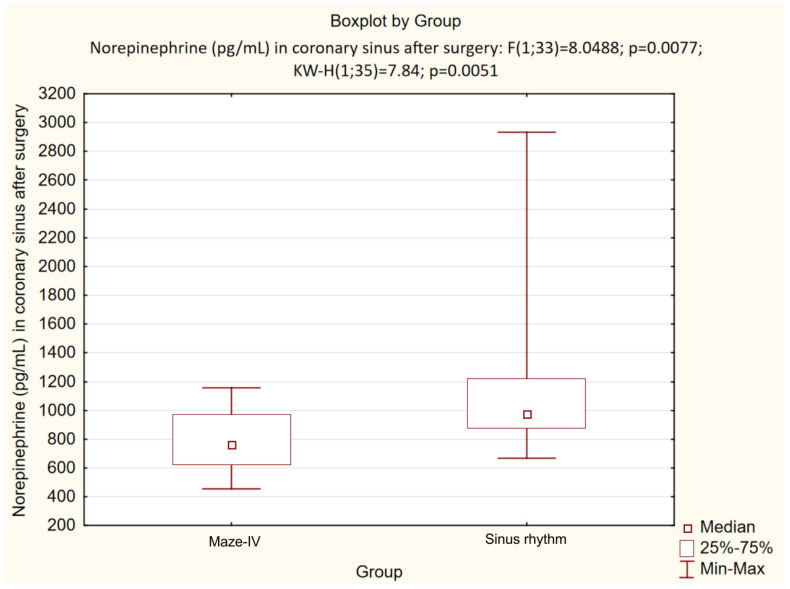
NE level in CS (pg/mL) after the main stage of surgery.

**Figure 3 life-13-01340-f003:**
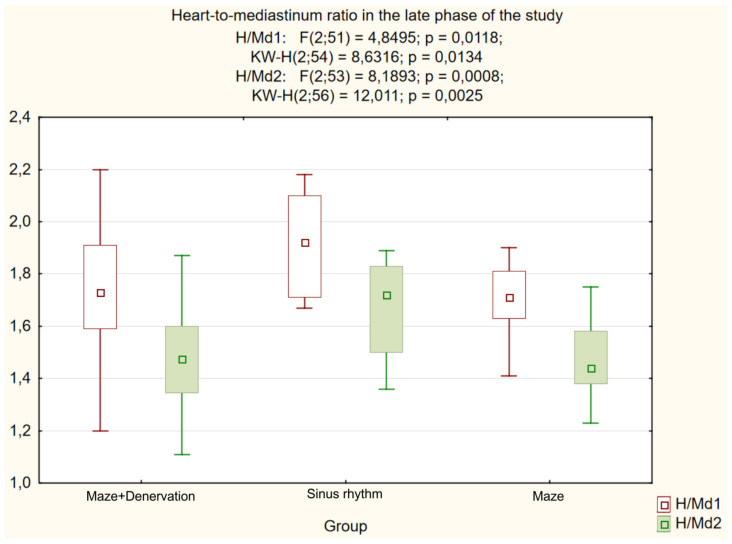
Heart-to-mediastinum ratio in the late phase of the study in the group with SR and AF.

**Figure 4 life-13-01340-f004:**
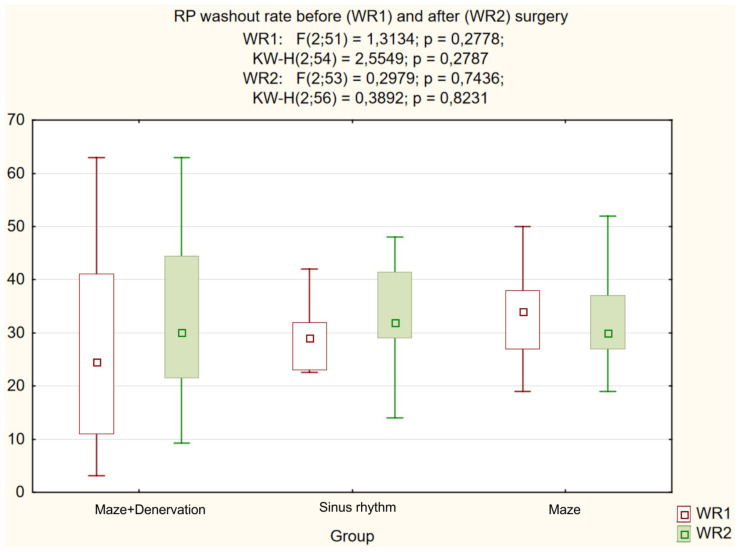
RP washout rate.

**Figure 5 life-13-01340-f005:**
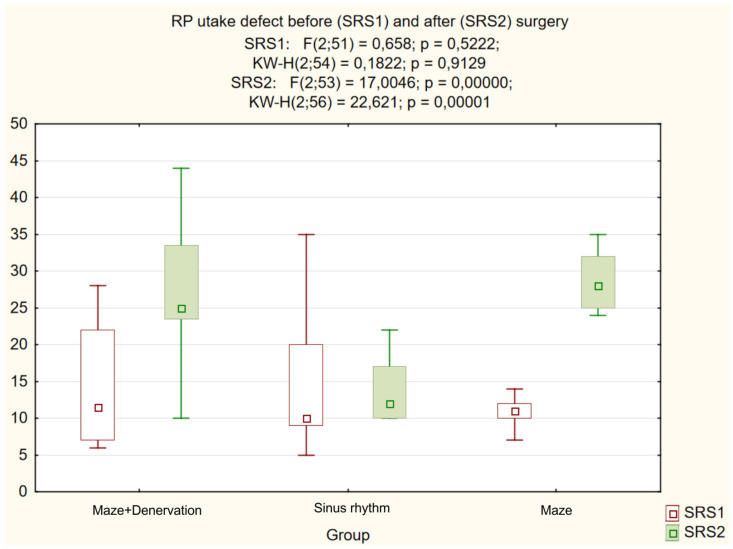
RP uptake defect.

**Figure 6 life-13-01340-f006:**
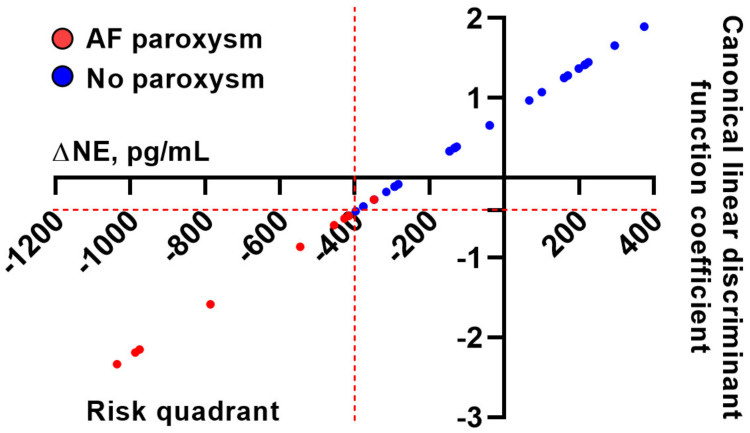
Distribution of the intraoperative difference between the levels of NE in the ascending aorta and in the coronary sinus (ΔNE), plotted against canonical linear discriminant function coefficient, across patients with and without AF paroxysm after RFA.

**Table 1 life-13-01340-t001:** Main preoperative clinical and instrumental parameters of patients with AF (group I) and SR (control group II), who underwent sympathetic nervous system examination (*n* = 68).

Parameter	Group I. AF: AHD Treatment + RFA (*n* = 53) Me (Q1; Q3)	Group II. Control Group with SR (*n* = 15) Me (Q1; Q3)	*p*-Value
Age (years)	59.5 (53; 64)	57.0 (53; 64)	0.91
Body mass index	31.1 (27.1; 34.3)	30.4 (26.4; 34.1)	0.21
Diabetes mellitus	5 (9.4%)	2 (13.3 %)	0.13
COPD	14 (26.4 %)	4 (26.6%)	0.41
Arterial hypertension	46 (87%)	14 (93 %)	0.39
Warfarin *	53 (100)	–	-
Amiodarone *	15 (28)	–	-
Beta-blockers *	44 (83.3)	8 (53.3)	0.03
Aspirin *	7 (13.3)	5 (33.3)	0.001
ACE inhibitors	27 (51)	15 (100)	0.02
Diuretics	50 (94.4)	4 (26.7)	0.004
Digoxin *	41 (77.7)	–	-
Calcium antagonists	12 (22.3)	5 (33.3)	0.02
LA diameter (mm)	52.0 (47; 55)	46.0 (40; 46)	0.002
LA volume (ml)	220.1 (170.3; 240.6)	131.2 (96.7; 142.2)	0.001
RV diameter (mm)	24.0 (21; 28)	22.0 (22; 24)	0.32
IVS (mm)	10.0 (9; 10)	9.0 (9; 10)	0.68
LVEDD (mm)	53.75 (51; 58)	53.0 (47; 56)	0.53
LVESD (mm)	33.5 (32; 39)	32.5 (29; 34)	0.19
LVEF (Simpson) (%)	66.0 (60; 68)	66.0 (65; 72)	0.31
LVEDV (mL)	132.5 (106; 194)	112.0 (104; 142)	0.63
LVESV (mL)	45.5 (35; 71)	49.0 (31; 50)	0.54
RVSP (mmHg)	44.0 (40; 50)	40.0 (38; 61)	0.93
EuroSCORE2, %	3.9 (2.9; 4.8)	3.2 (2.4; 4.5)	0.59
STS score	2.9 (2.3; 4.1)	3.6 (1.5; 5.5)	0.93
6 min walking test (m)	220.0 (200; 256)	322.0 (250; 356)	0.02

COPD—chronic obstructive pulmonary disease; ACE—angiotensin converting enzyme; LA—left atrium; RV—right ventricle; IVS—interventricular septum; LVEDD—left ventricular end-diastolic diameter; LVESD—left ventricular end-systolic diameter; LVEF—left ventricular ejection fraction; LVEDV—left ventricular end-diastolic volume; LVESV—left ventricular end-systolic volume; RVSP—right ventricular systolic pressure. * The drugs were canceled 7 days before the studies and surgery.

**Table 2 life-13-01340-t002:** Postoperative clinicopathological parameters of patients with AF (group I) and SR (control group II) who underwent sympathetic nervous system examination (*n* = 68).

Parameter	Group I. AF: AHD Treatment + RFA (*n* = 53) Me (Q1; Q3)	Group II. Control Group with SR (*n* = 15) Me (Q1; Q3)	*p*-Value
LA diameter before surgery (mm)	45.0 (44; 49)	39.0 (25; 41)	0.005
RV diameter before surgery (mm)	23.5 (20; 27)	20.0 (20; 22)	0.13
IVS before surgery (mm)	10.0 (9; 10)	9.0 (9; 10)	0.52
LVEDD before surgery (mm)	49.0 (47; 54)	46.0 (44; 48)	0.12
LVESD before surgery (mm)	33.0 (30; 37)	30.0 (26; 35)	0.18
LVEDV before surgery (mL)	119.5 (90; 127)	87.0 (72; 98)	0.24
LVESV before surgery (mL)	40.5 (34; 53)	31.0 (27; 37)	0.03
LVEF (Simpson biplane) before surgery (%)	60.5 (58; 66)	67.0 (64; 70)	0.08
RVSP before surgery (mmHg)	35.0 (30; 40)	35.0 (28; 37)	0.52
CPB time (min)	120.0 (110; 161)	120.0 (113; 140)	0.57
ACC time (min)	70.0 (57; 98)	92.0 (79; 110)	0.34
6 min walking test after surgery (m)	406.5 (380; 435.5)	412.0 (290; 450)	0.68

LA—left atrium; RV—right ventricle; IVS—interventricular septum; LVEDD—left ventricular end-diastolic diameter; LVESD—left ventricular end-systolic diameter; LVEF—left ventricular ejection fraction; LVEDV—left ventricular end-diastolic volume; LVESV—left ventricular end-systolic volume; RVSP—right ventricular systolic pressure; CPB—cardiopulmonary bypass; ACC—aortic cross-clamp.

**Table 3 life-13-01340-t003:** Assessment of the adequacy of the RF denervation efficacy model based on the Wilks’ Lambda criterion.

Parameter	Wilks’ Lambda: 0.56315 approx. F (3.23) = 5.9471. *p* < 0.0037					
	Wilks’ Lambda	Partial Lambda	F-Remove (1.23)	*p*-Value	Toler.	1-Toler. (R-sq.)
NE level gradient (Ao/CS, ΔNE, pg/mL)	0.89	0.64	13.20	0.001	0.99	0.01
Metanephrine level gradient (Ao/CS, ΔNM, pg/mL)	0.59	0.94	1.41	0.24	0.99	0.006
Normetanephrine level gradient (Ao/CS, ΔNNM, pg/mL)	0.59	0.94	1.34	0.25	0.99	0.005

**Table 4 life-13-01340-t004:** Classification matrix.

Group	The Percentage of Correct Assignments	G_1:1 *p* = 0.7037	G_2:2 *p* = 0.2963
G_1:1	94.7	50	3
G_2:2	75.0	3	12
Total	88.9	53	15

## Data Availability

The datasets used and analyzed during the current study are available from the corresponding author upon reasonable request.

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
