# Peer review of "Neurohumoral Markers of Cardiac Autonomic Denervation after Surgical Ablation of Long-Standing Persistent Atrial Fibrillation"

_life, 2023, doi:10.3390/life13061340_

Round 1

Reviewer 1 Report

The authors evaluate associates between RF during ablation on neurohumoral transmitter levels and 123I-MIBG uptake. The authors conclude that the technique may "predict the efficacy of the ‘Maze-IV’ procedure, which will allow assessing the risk of AF recurrence after ablation."

Assessment

The rationale of the study is clear, the studies appear to be well executes and designed, and the results are quite interesting.

Critique

0. The state aim of the study in the Abstract and elsewhere in the paper needs to be more clearly stated.  I would recommend, that the authors restate their aim as "to evaluate the consequences of radio frequency (RF) surgical ablation on both neurohumoral transmitter levels as well as myocardial iodine-123-metaiodobenzylguanidine (123I-MIBG) uptake in patients with of long-standing persistent atrial fibrillation (AF)."

1.  What is missing from the paper is an analysis of data related to the success rate AF termination/prevention following ablation and the relationship (or not) to neurohumoral transmitter levels and 123I-MIBG uptake.  This data is never evaluated (or discussed).  These results need to be explicitly presented and the stated conclusions need to directly incorporate these relationships.   Did denervation correlate with AF incidence or procedure success?

2.  The paper uses excessive jargon and the authors need to define some of their measures and/or terms used in the paper.  For example (page 6), what is "sympathetic nerve system tonus"?  What is being compared and measured in relationship to this statement?  Also (page 7), what is an "uptake defect"?  What is being compared and measured?  What do the authors mean by the term "the efficacy of AF RF ablation" (page 7)? Efficacy of what?  What are "levels of vegetative regulation"?  This is not a useful, common or meaningful term.  

3.  The Discussion is far too long.  Moist of it has nothing to do with the measurements.  The authors need to eliminate ~70% of this discussion and focus rather on provide a meaningful discussion of their results and the limitations.  In particular, I would be far more interested in the authors thoughts on how autonomic nerve parameters correlated (or not) with successful ablation.

4. The drug status/therapies of the patients should be included in Table 1.

5.  The data in all Figures needs to display the individual patient measurements.

6.  I always get concerned when I see unexpected patterns of data presentation.  For example, Table 1 presents multiple measures of patients using 1 significant digit (i.e. decimal place).  It seems unlikely (virtually impossible actually) for 16 of 21 of these measures to have a value of zero.  How do the authors explain this? Unexpected patterns also appear in Table 2.

Author Response

We sincerely thank the reviewer for the constructive criticism and valuable notes, which collectively helped us to improve the paper. Please see the attachment.

Reviewer 2 Report

The manuscript entitled: Neurohumoral Markers of Cardiac Autonomic Denervation after Surgical Ablation of Long-standing Persistent Atrial Fibrillation is an original article. The theme is very interesting because it refers to long-standing persistent atrial fibrillation in patients underwent surgical valve correction. The authors present a randomized prospective study. The theme is important.

I have some comments supporting the decision of major revision.

The introduction is too long (for example data about AF). I recommend replace data about AF with those about myocardial iodine-123-metaiodobenzylguanidine and denervation to better understand the methodology of this study (page 5 lines 167-172).

In table 1:

1.      I recommend putting parameters  as following: clinical (including gender, BMI, main comorbidities), echocardiographic and scores (Euroscore 2; please add STS score).

2.      LVEF by Teicholz method is no more recommended by the current guidelines especially because in this study there are patients with abnormal wall kinetics.  Therefore, I recommend not putting in results.

3.      LA diameter is no more recommended for assessing LA size; LA area or LA volume is now recommended especially in LA structural remodeled. Please add these parameters.

How was detected sinus rhythm? By ECG or by Holter monitoring? It is mandatory to specify how and in what moment of this study. Otherwise, all methodology is debatable.

I disagree with this: …``results of 6-minute walking test that was most likely due to AF.`` Please putt the comorbidities of these patients.

The authors said: …``the significance (p = 0.001) of the aortic root and coronary sinus gradient of NE level 0.405 pg/mL was determined as a cut-off value for efficacy evaluation of the RF denervation.`` Please explain what is the scientific base of this statement.

Study limitations are poor (taking into consideration the above comments).

Please reconsider the conclusions according with the results. There are many assumptions and dependencies.

References are very few. There are other references with an important message related to this subject.

English language has some errors.

Author Response

(The authors gave the same response as above.)

Round 2

Reviewer 2 Report

Unfortunately the authors did not responded to all my recommendations.

The authors responded: "We actually agree with the reviewer but we believe that the entire introduction contains important data necessary for understanding the problem, so we would not like to shorten it. If You insist, we will try to do it."

There are phrases, which might be eliminated or restructurated. For example:

"The autonomic nervous system (ANS) plays an important role in modulation of cardiac electrophysiology and arrhythmogenesis [3]. Disorders of the ANS components can be a prognostic factor of an unfavorable course of cardiovascular diseases. Many studies have contributed to a better understanding of the anatomy and physiology of the ANS and have provided evidence of the relationship between ANS and clinically significant arrhythmias [3]. They demonstrated the prospects for the development of this area of research in arrhythmology."

Another important issue is the fact that references are not consecutively. For example, the above paragraph is the first and begin with reference number 3.

The authors repeatedly contradict the comments made. The remarks as: ``In our clinic, this indicator is not calculated, and some authors point out that it does not reflect the risk of mortality.`` are not appropriate as the current surgical guidelines recommend them. This is not allowed to be affirmed as researcher.

3. I do not ask to remove LA diameter. My recommendation was: ``Please add these parameters``.   The authors said: ``However, if You insist, we will remove it.`` This formulation is repeated severely times.  

Comment number 5: I disagree with this: …``results of 6-minute walking test that was most likely due to AF.``

The authors do not understand that the results of 6-minute walking test depend also on the comorbidities and there is not correct this statement.

Comment number 6: The authors responded:  "In our opinion, this is explained in sufficient detail on pages 10 and 11. In accordance with the reviewer’s suggestion, we have added an additional comment after figure 6 (highlighted in yellow in the text of the revised manuscript, lines 308-315)."

After they underlined that ‘’this is explained in sufficient detail’’ they added, however, 7 lines. I would say that this is not a scientific attitude.

Comment number 8: ‘’ We agree with the reviewer that conclusions were vaguely articulated in the initial manuscript. We have changed the conclusion in accordance with your recommendations (highlighted in yellow in the text of the revised manuscript, lines 397-399). We decided to delete the fifth conclusion.’’

This comment of the authors underlined very well that your conclusions are not in accordance with the scientific rigor. I could not accept this type of comments as reviewer. This manuscript might be reedited according with the scientific rigor.

The authors added 6 references and now the manuscript have 22 references, which there are still fewer. 

In conclusion, the authors do not responded rigorously to all my suggestions/comments. The manuscript have still many issues and in my opinion is not readily to be accepted.

Author Response

(The authors gave the same response as above.)

Round 3

Reviewer 2 Report

Thank you for responding to my comments.